

# Asiatic acid reduces lipopolysaccharides-induced pulp inflammation through activation of nuclear factor erythroid 2-related factor 2 in rats

Risya Cilmiaty[1,2], Arlina Nurhapsari[3], Adi Prayitno[2], Annisa Aghnia Rahma[4] and Muhana Fawwazy Ilyas[4]

[1] Doctoral Program of Medical Sciences, Faculty of Medicine, Universitas Sebelas Maret, Surakarta, Central Java, Indonesia
[2] Department of Oral Diseases, Faculty of Medicine, Universitas Sebelas Maret, Surakarta, Central Java, Indonesia
[3] Department of Conservative Dentistry, Faculty of Dentistry, Islamic University of Sultan Agung, Semarang, Central Java, Indonesia
[4] Medical Profession Program, Faculty of Medicine, Universitas Sebelas Maret, Surakarta, Central Java, Indonesia

Corresponding author
Risya Cilmiaty,
risyacilmiaty@staff.uns.ac.id

## ABSTRACT

**Background:** Dental pulp inflammation, often initiated by Gram-negative microorganisms and lipopolysaccharides (LPS), can lead to pulpitis and, subsequently, dental pulp necrosis, compromising tooth structure and increasing susceptibility to fracture. Asiatic acid, derived from *Centella asiatica*, has demonstrated pharmacological properties, including anti-inflammatory and antioxidant effects, making it a potential candidate for mitigating LPS-induced pulp inflammation. This *in vivo* study aims to investigate the impact of Asiatic acid on the nuclear factor erythroid 2-related factor 2 (Nrf2) pathway in *Rattus norvegicus* with LPS-induced pulp inflammation.

**Methods:** This quasi-laboratory experimental *in vivo* study employed a post-test-only control group design to investigate the effects of Asiatic acid on LPS-induced pulp inflammation in Wistar rats. Thirty rats were randomly divided into six groups subjected to various interventions. LPS was administered to all groups for 6 h except the standard control group (CG, $n = 5$). The negative control group (NCG, $n = 5$) received only glass ionomer cement. The positive control group (PCG, $n = 5$) received Eugenol with glass ionomer cement. Intervention groups 1, 2, and 3 (IG1, IG2, IG3; $n = 5$ each) received Asiatic acid at concentrations of 0.5%, 1%, and 2%, respectively, with glass ionomer cement. Dental pulp inflammation was confirmed through immunological (tumor necrosis factor alpha (TNF-α) levels), histopathological (inflammatory parameters), and physiological (pain assessment using the rat grimace scale) analyses. Additionally, Nrf2 levels were examined using enzyme-linked immunosorbent assay (ELISA).

**Results:** Asiatic acid administration significantly influenced Nrf2 levels in rats with LPS-induced pulp inflammation. Nrf2 levels were significantly higher in groups treated with 0.5% (IG1) ($8.810 \pm 1.092$ ng/mL; $p = 0.047$), 1.0% (IG2) ($9.132 \pm 1.285$ ng/mL; $p = 0.020$), and 2.0% (IG3) ($11.972 \pm 1.888$ ng/mL; $p = 0.000$) Asiatic

acid compared to NCG (7.146 ± 0.706). Notably, Nrf2 levels were also significantly higher in the 2.0% Asiatic acid group (IG3) compared to the PCG treated with Eugenol (8.846 ± 0.888 ng/mL; $p = 0.001$), as well as IG1 ($p = 0.001$) and IG2 ($p = 0.002$). However, no significant difference was observed between administering 0.5% Asiatic acid (IG1), 1.0% Asiatic acid (IG2), and Eugenol (PCG).

**Conclusion:** This research showed that Asiatic acid significantly impacted the Nrf2 levels in rats with LPS-induced pulp inflammation. This suggests that it has the potential to be used as a therapeutic agent for reducing dental pulp inflammation. These findings support the need to further explore Asiatic acid as a promising intervention for maintaining dental pulp health.

# INTRODUCTION

The dental pulp comprises connective tissue, nerve cells, blood vessels, and various types of cells that play specific roles in supporting the tooth's normal function. Rat models are commonly used in dental research to study treatments for dental pulp, as their dental structures and cell functions are similar to those in humans (*Huang et al., 2023*). Inflammation of the dental pulp is a complex process involving nerve, blood, vessel, and immune system responses. It is mainly caused by certain types of bacteria known as Gram-negative microorganisms. These bacteria produce lipopolysaccharides (LPS), which is found in their outer membrane and plays a significant role in triggering inflammation in the dental pulp (*Brodzikowska et al., 2022*). If pulpitis, which is inflammation of the pulp, is not treated, it can lead to the death of the dental pulp. This might weaken the tooth structure, making it more prone to fractures. Therefore, it is essential to maintain the vitality of the pulp and treat pulpitis to prevent dental pulp necrosis for effective tooth function (*Colombo et al., 2014*).

Plant extracts have demonstrated significant pharmacological properties and potential for treating various medical conditions (*Ghozali et al., 2023*; *Novika et al., 2024*; *Geszke-Moritz, Nowak & Moritz, 2023*; *Capasso & Di Cesare Mannelli, 2020*). Asiatic acid isolates, a saponin (triterpenoids) component extracted from *Centella asiatica*, have received significant attention considering their pharmacological features and potential for treatment in various medical conditions (*James & Dubery, 2009*; *Kamble, Goyal & Patil, 2014*; *Kamble et al., 2017*). Asiatic acid exhibits various pharmacological uses, including anti-inflammatory, antioxidant, antinociceptive, antimicrobial, and anticancer properties (*Polash et al., 2017*; *Ogunka-Nnoka et al., 2020*). This isolate has been investigated for its potential modulatory effects on the nuclear factor erythroid 2-related factor 2 (Nrf2) pathway as a defense mechanism, particularly in inflammation, by inhibiting oxidative stress (*Kamble et al., 2017*). Activation of Nrf2 is believed to potentially increase antioxidant and cytoprotective genes, aiding in the reduction of cellular damage. Therefore, this *in vivo* study aims to examine the impact of Asiatic acid on Nrf2 in *Rattus norvegicus* with LPS-induced pulp inflammation.

## MATERIALS AND METHODS

### Study design

This quasi-laboratory experimental *in vivo* study uses a post-test-only control group design approach. This study was conducted at the Experimental Animal Handling Laboratory and Molecular Biology Laboratory, Faculty of Medicine, Islam Sultan Agung University, Semarang, Indonesia, in August 2022. The protocol of this study has been registered and approved by the Health Research Ethics Commission, Faculty of Medicine, Gadjah Mada University (UGM), Yogyakarta, Indonesia, with registration number KE/FK/0703/EC/2020 on 29 June 2020. All methods followed the relevant guidelines and regulations for the welfare of UGM laboratory animals. This study also confirmed the Animal Research: Reporting of *In Vivo* Experiments (ARRIVE) guidelines (*Percie du Sert et al., 2020*). To ensure unbiased results, allocation, the conduct of the experiment, and the outcome assessment were carried out by blind laboratory assistants.

### Study subject

The study subjects were white rats (*Rattus norvegicus*) of the Wistar strain retrieved from the Pharmacology Laboratory, Faculty of Medicine, UGM, Yogyakarta, Indonesia. The criteria include male, age 8–10 weeks, body weight (BW) 200–250 g, and in healthy condition without anatomical abnormalities or physical defects. Exclusion criteria for this study included rats that had contracted a disease or died during the study period, were unable to adapt to the environment, or had experienced a weight loss of more than 10% during the adaptation period. The animals were sourced from a reputable supplier, Kemuning (CV. Dunia Kaca), Karanganyar, Indonesia, with a certificate of cultivation number of 524/082.19/I/2019, after thorough health assessments to ensure optimal health and immune status. Only animals with confirmed wild-type genotypes and no previous procedures were included in the study, enhancing the reliability of the experimental data. This study's sample size was determined using a power analysis of the mean results of preliminary research with a significance level of 0.05 and a power of 0.80.

Thirty rats were grouped by computer-generated randomization into six groups. LPS induced the entire group for 6 h except the standard control group (CG) ($n = 5$). Subsequently, the negative control group (NCG) was given glass ionomer cement only ($n = 5$); the positive control group (PCG) was given Eugenol using paper point + glass ionomer cement ($n = 5$); intervention group 1 (IG1) was given Asiatic acid 0.5% using paper point + glass ionomer cement ($n = 5$), intervention group 2 (IG2), was given Asiatic acid 1% using paper point + glass ionomer cement ($n = 5$), intervention group 3 (IG3), Asiatic acid 2% using paper point + glass ionomer cement ($n = 5$). The intervention groups were divided into three groups to investigate each result from several percentage dosages of Asiatic acid.

### Treatment procedure

Before the treatment, rats were acclimated for more than a week. Animal care, feeding, housing, and enrichment were carried out as previously outlined in *Nurhapsari et al. (2023)*. The maxillary incisor teeth of Wistar rats were prepared with a low-speed stainless

steel round bur 0.10 to a depth of ±5 mm until they reached the pulp roof; each rat tooth sample was applied with LPS (20 mg/ml) using a paper point for 6 h in the cavity of the maxillary incisor teeth so can result in pulp inflammation. Next, the teeth were filled with GIC Fuji VII. After 6 h of LPS administration, the cavities in groups PCG, IG1, IG2, and IG3 were opened, and paper points were taken, while in NCG, the pulp tissue was immediately taken. The tooth was split to obtain pulp tissue, extirpation of the pulp in the tooth was carried out with a #40 barbed broach, then the pulp tissue was washed in a petri dish containing NaCl and stored in a microtube at −20 °C. Pulpitis was created in the treatment group (NCG, PCG, IG1, IG2, and IG3) by preparing the maxillary incisors and giving LPS for 6 h. After 6 h of administration of LPS, treatment was carried out in groups of PCG (Eugenol using paper point + glass ionomer cement), IG1 Asiatic acid 0.5% using paper point + glass ionomer cement), IG2 (Asiatic acid 1% using paper point + glass ionomer cement), and IG3 (Asiatic acid 2% using paper point + glass ionomer cement). Analgesia or anesthesia was not administered to prevent interference with measurements and to preserve the integrity of the experimental model, as pain responses served as physiological confirmatory indicators of the animal model. Euthanasia criteria were established to ensure the humane termination of animals before the planned end of the experiment. This was deemed necessary to minimize suffering and distress. Rats were sacrificed after 72 h of treatment, and the maxillary incisors were removed. The surviving animals were euthanized using $CO_2$ asphyxiation following the American Veterinary Medical Association (AVMA) guidelines. No animals died before the end of the experiment or before they could be humanely euthanized. The treatment procedure is illustrated in Fig. 1.

## Confirmation of animal model

To confirm dental pulp inflammation in an animal model, various methods were used, including immunological, histopathological, and physiological analyses. The immunological analysis involved examining the level of tumor necrosis factor alpha (TNF-α). This was done by measuring the TNF-α level in the supernatant of pulp tissue using a TNF-α enzyme-linked immunosorbent assay (ELISA) kit (Rat TNF-α, BZ-08184670-EB, Bioenzy) in accordance with the manufacturer's protocols (*Nurhapsari et al., 2023*). Subsequently, histopathological analysis was also performed. After the deparaffinization process with xylene, the tissue slice slides were transferred to an aqueous medium by decreasing alcohol levels. Then, the slides were washed with running water, placed in hematoxylin paint for 7–10 min, and washed again with running water. Next, the slide was placed in eosin paint for 2 min, washed with running water, and rinsed with 90% alcohol. Finally, the slides were air-dried, cleared with xylene, and covered with a cover slip. Observations were carried out with a microscope magnification of 400 times in three fields of view with an area of $17 \times 10^{-5}$ mm$^2$ for each glass object. Parameters and scores used for histopathological evaluation of pulp tissue (*He et al., 2017*) include the location of inflammatory cells, intensity of inflammatory infiltrate, edema, vascular leakage, and necrosis. Last, a physiological analysis was carried out using pain assessment. During the

**Figure 1 Diagram showing the procedure of the LPS-induced inflammation and the treatment of Eugenol or Asiatic acid.** Created with Canva and vectr.com.

study period, rat pain symptoms were noted, as were rat pain scores using the rat grimace scale (RGS) (*Sotocina et al., 2011*).

## Examination of Nrf2 level

The levels of Nrf2 were measured using an Nrf2 ELISA kit (Rat Nrf2, BZ-08183801-EB, Bioenzy) according to the manufacturer's protocols. The tissue was sonicated and centrifuged to obtain the supernatant, and an ELISA test was carried out using the sandwich technique to see the levels of Nrf2 in each group. The process involves a plate filled with Nrf2 antibodies that will bind to the antigen in the sample. Streptavidin-HRP is given after the Nrf2 antigen is placed in the well. Then, washing and adding substrate were carried out to see Nrf2 levels *via* ELISA reader (*Sakamoto et al., 2018*).

## Statistical analysis

Descriptive analysis was used to determine the data distribution and concentration. The Shapiro-Wilk test was performed to determine the distribution of the data. Levene's test was used to assess the homogeneity of the data between groups. The differences between groups were analyzed using an independent T-test for TNF-$\alpha$ level, Pearson Chi-square test for histopathological analysis, Mann-Whitney test for RGS score, and one-way ANOVA + *post-hoc*-LSD for Nrf2 level. All statistical tests were two-sided, and *P*-values of <0.050 were considered statistically significant. Statistical analyses were performed using IBM SPSS Statistics for Windows (version 24.0; IBM Corp. Armonk, NY, USA).

# RESULTS

## Animal model of LPS-induced pulp inflammation

This study was carried out on 30 rats. No rats or tooth samples were excluded or dropped out in this study. In this study, the creation of an animal model of dental pulp inflammation was successfully proven by immunological, histopathological, and physiological analyses. Immunological analysis, there was a significant increase in TNF$\alpha$ level ($p$ = <0.01) on NCG (175.82 ± 3.87 ng/mL; $p$ = 0.015) compared to CG (117.02 ± 27.37 ng/mL) in the pulp tissue. Furthermore, there are also differences in histopathological analysis between the location of inflammatory cells ($p$ = 0.002), intensity of inflammatory infiltrate ($p$ = 0.007), and vascular leakage ($p$ = 0.038) between NCG and CG. Meanwhile, the two groups had no difference in edema and necrosis. Last, physiological analysis showed a significant difference in the rat grimace scale for NCG

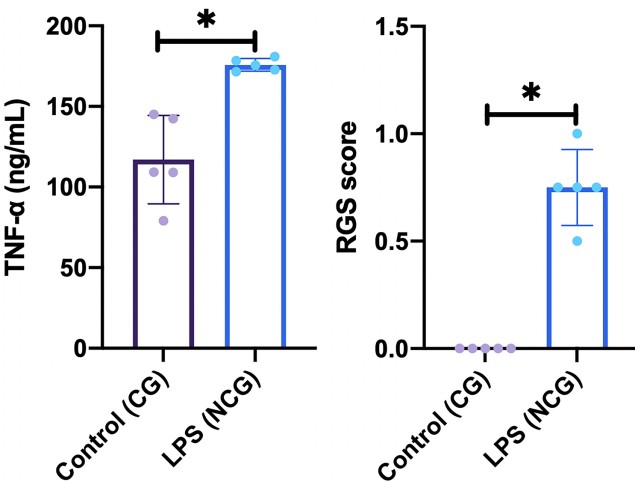

**Figure 2 Result of immunological (TNF-α level) and physiological (rat grimace scale examination) analysis on the animal model.** Data are presented as mean and standard deviation as well as individual values. The differences between groups were analyzed using an independent T-test for TNF-α level and Mann-Whitney for RGS score. An asterisk (*) is considered statistically significant ($p < 0.050$). CG, standard control group; LPS, lipopolysaccharides; NCG, negative control group: RGS, rat grimace scale; TNF-α, Tumor necrosis factor alpha.

$(0.75 \pm 0.18; p = 0.008)$ compared to CG $(0.00 \pm 0.00)$. Data from the immunological and physiological analysis are presented in Fig. 2, and data from histopathological analysis are shown in Table 1 and Fig. 3 (*Nurhapsari et al., 2023*).

### Effect of Asiatic acid on Nrf2 level

This study found that administration of Asiatic acid significantly affected Nrf2 levels in *Rattus norvegicus* with LPS-induced pulp inflammation (visualized in Fig. 4). The Nrf 2 level was significantly higher in the groups given Asiatic acid of 0.5% (IG1) $(8.810 \pm 1.092$ ng/mL; $p = 0.047)$, 1.0% (IG2) $(9.132 \pm 1.285$ ng/mL; $p = 0.020)$, and 2.0% (IG3) $(11.972 \pm 1.888$ ng/mL; $p = 0.000)$ when compared with NCG $(7.146 \pm 0.706)$. Furthermore, the Nrf2 level was also significantly higher in the group given 2.0% Asiatic acid (IG3) when compared with the group given Eugenol (PCG) $(8.846 \pm 0.888$ ng/mL; $p = 0.001)$ and also Asiatic acid of 0.5% (IG1) $(p = 0.001)$ and 1.0% (IG2) $(p = 0.002)$. However, it was found that there was no significant difference between administering Asiatic acid of 0.5% (IG1), 1.0% (IG2), and Eugenol (PCG) $(p > 0.05)$.

## DISCUSSION

### Animal model study of dental pulp inflammation

Pulp inflammation is a complex process involving neural, vascular, and immune system responses typically induced by Gram-negative microbes. LPS are elements that contribute substantially to the pathogenesis of inflammation, including pulpitis (*Brodzikowska et al., 2022*). Pulpitis starts through a particular injury that produces mediators such as chemokines and cytokines to attract immune cells, including macrophages and neutrophils, into the inflammatory areas (*Germolec et al., 2018*). The immediate

**Table 1  Histological description of pulp tissue on animal model.**

| Parameter | Score | CG | NCG | p-value |
|---|---|---|---|---|
| Location of inflammatory cells | 1 (None) | 5 (100%) | 0 (0%) | 0.002* |
| | 2 (Limited to exposed locations) | 0 (0%) | 5 (100%) | |
| | 3 (Up to the odontoblast layer) | 0 (0%) | 0 (0%) | |
| | 4 (Scattered throughout the coronal pulp) | 0 (0%) | 0 (0%) | |
| Intensity of inflammatory infiltrate | 1 (0–20 inflammatory cells) | 5 (100%) | 0 (0%) | 0.007* |
| | 2 (21–40 inflammatory cells) | 0 (0%) | 4 (80%) | |
| | 3 (41–80 inflammatory cells) | 0 (0%) | 1 (20%) | |
| | 4 (>80 inflammatory cells) | 0 (0%) | 0 (0%) | |
| Oedema | 1 (None) | 5 (100%) | 4 (80%) | 0.292 |
| | 2 (Available) | 0 (0%) | 1 (20%) | |
| Vascular leakage | 1 (None) | 5 (100%) | 2 (40%) | 0.038* |
| | 2 (Available) | 0 (0%) | 3 (60%) | |
| Necrosis | 1 (None) | 5 (100%) | 4 (80%) | 0.292 |
| | 2 (Available) | 0 (0%) | 1 (20%) | |

Notes:
Data are presented as frequency (and percentage). The differences between groups were analyzed using the Pearson Chi-square test.
* Considered statistically significant ($p < 0.050$).
CG, standard control group; NCG, negative control group.

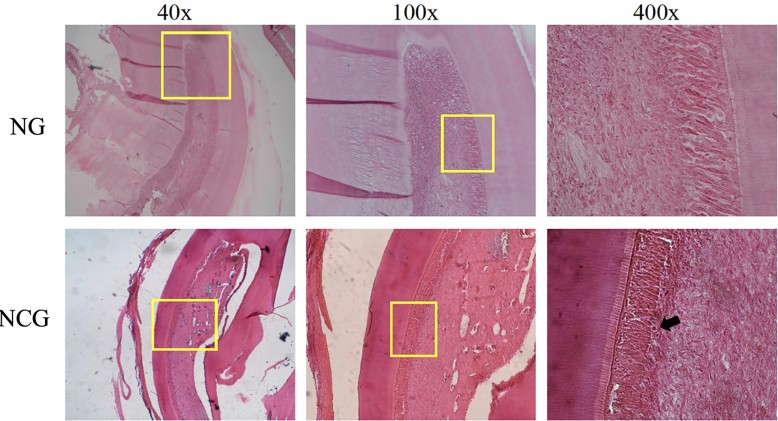

**Figure 3  Histological description of pulp tissue on the animal model.** Images were taken using magnifications of 40, 100, and 400 times. The yellow box shows the section under magnification, the black arrow shows inflammatory cells, and the yellow arrow shows vascular leakage which is characterized by the release of erythrocytes from the blood vessels. CG, standard control group; NCG, negative control group.

inflammatory response is regulated by various kinds of molecules, notably toll-like receptors (TLRs) and reactive oxygen species (ROS) leading to oxidative stress and inflammation (*Lu, Yeh & Ohashi, 2008*; *Brenner et al., 2014*; *Landén, Li & Ståhle, 2016*).

On the other hand, ROS will catalyst the mediator-signaling molecules, which includes the NF-κB pathway, thereby up-regulates the synthesis of pro-inflammatory chemicals, including TNF-α (*Buelna-Chontal & Zazueta, 2013*; *Nurhapsari et al., 2023*).

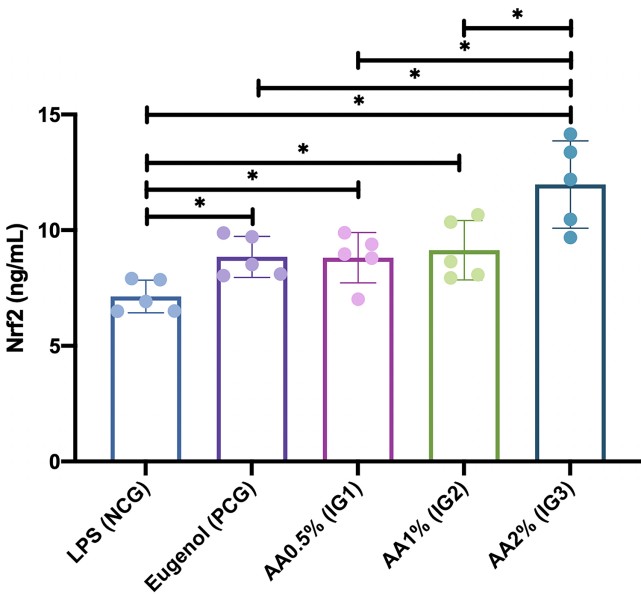

**Figure 4 Nrf2 levels after intervention in each group in *Rattus norvegicus* with LPS-induced pulp inflammation.** Data are presented with mean and standard deviation as well as individual values. The differences between groups were analyzed using One-way ANOVA + *post-hoc*-LSD. An asterisk (*) is considered statistically significant ($p < 0.050$). AA, Asiatic acid; IG, Intervention Group; LPS, lipo-polysaccharides; CG, standard control group; NCG, negative control group; PCG, positive control group.

Moreover, this study observed a significant increase in TNF-α levels between NCG and CG. This finding aligns with previous theories regarding LPS-induced oxidative stress and the immune system, specifically TNF-α. Oxidative stress may enhance the production of TNF-α from immune cells, particularly macrophages, and TNF-α, in turn, can contribute to further oxidative stress.

Upon encountering LPS, immune cells such as macrophages and dendritic cells recognize it and initiate intracellular signaling pathways, including activating NF-κB, a pivotal transcription factor in inflammation. NF-κB translocates into the nucleus and binds to the promoter region of the TNF-α gene; once synthesized, TNF-α is released into the extracellular space. TNF-α binds to its receptors on immune cells and endothelial cells, initiating a cascade of inflammation such as cytokine production, leukocyte recruitment, and endothelial activation, leading to detrimental effects, including tissue damage and inflammation such as pulpitis (*van der Bruggen et al., 1999*; *John et al., 2008*).

The histology of tissues in LPS-induced inflammation can show characteristic changes associated with the inflammatory response that might vary depending on the tissue type and the duration of exposure to LPS. We recognized the shift in histological characteristics in this study, including the location of inflammatory cells, the intensity of inflammatory infiltration, and vascular leakage in the affected tissue. Inflammation typically involves the infiltration of immune cells into the affected tissue. This can include neutrophils, macrophages, and other immune cells (*Page, Kell & Pretorius, 2022*). These cells migrate to the site of inflammation to combat the perceived threat which later can lead to tissue

damage. This damage may be reflected in histological examination by disrupting normal tissue architecture or other structural abnormalities (*Lin et al., 2015*; *Li et al., 2021*). Inflammation may also cause vascular leakage through several mechanisms involving endothelial barrier integrity and function alterations. Endothelial cells line the inner surface of blood vessels and play a crucial role in maintaining vascular integrity and regulating the passage of fluids, solutes, and cells between the bloodstream and surrounding tissues. During inflammation, various inflammatory mediators released by activated immune cells and injured tissues can disrupt the endothelial barrier, leading to increased vascular permeability and leakage.

Moreover, inflammation and neuropathic-related pain have been evaluated using RGS (*Domínguez-Oliva et al., 2022*; *Nurhapsari et al., 2023*). It is a method for assessing pain by examining facial expressions and features, such as orbital tightening, nose/cheek flattening, and ear and whisker changes. An increase in the RGS indicates pain or discomfort in the rats following the administration of LPS. The facial expressions captured by the RGS serve as a behavioral indicator of pain. It suggests that the inflammatory response induced by LPS causes pain or discomfort in animals.

Inflammation can cause pain due to the release of various inflammatory mediators that sensitize nerve endings and heighten their response to stimuli. During inflammation, immune cells release prostaglandins, bradykinin, histamine, and cytokines like TNF-α and interleukins. These molecules activate nociceptors and generate action potentials transmitted to the central nervous system (CNS), resulting in pain perception (*Fang et al., 2023*).

In conclusion, exposure to LPS triggers pulpitis in our experimental animals. The changes in immunological analysis, including TNFα and histopathological and physiological analysis, may indicate the effect of LPS on the dental pulp's inflammatory response.

## Effect of Asiatic acid on Nrf2 level

Various compounds and materials have been used to treat pulp inflammation, each with its mechanisms of action and associated disadvantages (*Qureshi et al., 2014*; *Meschi, Patel & Ruparel, 2020*; *Cilmiaty & Ilyas, 2024*). *Centella asiatica*'s therapeutic effects as an antibacterial, antioxidant, and anti-inflammatory are strongly correlated with the formation and amounts of several secondary metabolites (*Polash et al., 2017*). Among these substances, triterpene saponins, particularly Asiatic acid, represent the primary metabolites implicated in *Centella asiatica*'s biochemical activity (*Shen et al., 2019*; *Ren et al., 2021*). In this study, our results showed that Asiatic acid, as an effective compound, could attenuate inflammation by the Nrf2 activation. We also recognized that the therapeutic effect of Asiatic acid concentration to increase Nrf2 was started in 0.5% and 1% concentration with the optimal dose of 2% concentration in the pulpitis-induced model.

Several mechanisms contribute to the observed findings. During the pathological inflammatory process induced by LPS, various immune cells, such as monocytes, macrophages, and lymphocytes, are initially activated. The cells further proceed to migrate toward the area of injury, which leads to the production of ROS and affects DNA. These

pro-inflammatory cells simultaneously release enormous quantities of pro-inflammatory mediators including prostaglandins, chemokines, and cytokines. These mediators will later attract macrophages into inflammation sites and consequently engage numerous transduction and transcription pathways that are responsible for inflammation, including Nrf2 (*Kaulmann & Bohn, 2014*; *Ahmed et al., 2017*).

In response to oxidative stress, human cells have established protective strategies that prevent the production of ROS by modulating Nrf2 signaling (*Kovac et al., 2015*; *Qi et al., 2017*). Nrf2 is crucial for overcoming oxidative stress. In physiological settings, the inactive Nrf2 is attached to Kelch-like ECH-associated protein 1 (Keap1) in the cell's cytoplasm. Under particular circumstances, notably oxidative stress, Nrf2 gets released from the Nrf2-Keap1 complex and transported to the nucleus (*Kensler, Wakabayashi & Biswal, 2007*). In the nucleus, Nrf2 will be linked to the antioxidant response element (ARE) (*Ma, 2013*).

Furthermore, Asiatic acid initiated the Nrf2 signal, which is strongly linked to promoting Nrf2 nuclear translocation, lowering Keap1 expression, and enhancing ARE activity. Previous research has revealed that Nrf2 signal amplification enhanced the expression of antioxidant genes involving nicotinamide adenine dinucleotide phosphate (NADPH), heme oxygenase-1 (HO-1), and other particles that protect cells from various injuries *via* their anti-inflammatory effects (*Kovac et al., 2015*; *Qi et al., 2017*).

As seen in this study's findings, Asiatic acid is widely recognized for its pivotal role in suppressing oxidative stress, thereby improving Nrf2 production (*Nurhapsari et al., 2023*). Nrf2 initiates the HO-1 gene and suppresses NFκB signaling. The Nrf2/HO-1 axis 1 regulates LPS-induced inflammatory responses. The activation of Nrf2 diminished the foam cell macrophage phenotype and inhibited excessive macrophage inflammation. Increased HO-1 expression *via* the Nrf2 pathway shields cells against death, demonstrating their potential utilization on behalf of inflammatory diseases.

While faced with oxidative stress, pro-inflammatory cytokines, including IL-6 and IL-1β, are excessively produced, triggering damage in target cells (*Ahmed et al., 2017*). Nrf2 activated by Asiatic acid reduces the formation of downstream IL-17 and other inflammatory substances, including Th1 and Th17, and prevents the expression of the mentioned genes induced by LPS. This condition subsequently activates NF-κB and leads to increased cytokine production. Initiation of the Nrf2/ARE pathway is critical in interrupting the cycle. Elevated Nrf2 lowers the synthesis of pro-inflammatory cytokines and chemokines and reduces NF-κB activity. Nrf2 regulates COX-2, IL-113, IL-6, and TNF-α, reducing the inflammation and damage. The results imply that Nrf2 represents an essential modulator for both critical cytoprotective mechanisms: anti-inflammation and anti-oxidation (*Ahmed et al., 2017*). Asiatic acid promotes PPAR-γ, limiting LPS-induced NF-κB activation and inflammatory mediator production such as PGE2, NO, IL-6, and IL-8 (*Hao et al., 2017*).

Overall, the previously mentioned experimental models proved that the Nrf2/HO-1 axis is essential in anti-inflammatory function, indicating that Nrf2 is a potential therapeutic target in inflammation-related disorders, including pulpitis.

### Limitation of study

This study is limited by its use of only male white rats of the Wistar strain, which may restrict the applicability of the findings to other populations or genders. Additionally, the study only evaluated the effects of Asiatic acid isolate on specific markers. Further research may be necessary to understand its broader impact on dental pulp inflammation. Moreover, the 72-h duration of the study may not capture the long-term effects of the interventions, and longer observation periods could offer a more comprehensive understanding of the outcomes.

## CONCLUSIONS

The Asiatic acid isolate has potential therapeutic benefits for treating dental pulp inflammation induced by lipopolysaccharide. This study found that Asiatic acid could reduce inflammation by increasing Nrf2 levels at concentrations of 0.5% and 1%, with the optimal dose being 2%. The increased activation of Nrf2 by Asiatic acid was linked to enhanced ARE activity and the expression of antioxidant genes, indicating its potential as a therapeutic target for inflammation-related disorders such as pulpitis.

### Funding

The authors received no funding for this work.

### Competing Interests

The authors declare that they have no competing interests.

### Author Contributions

- Risya Cilmiaty conceived and designed the experiments, authored or reviewed drafts of the article, and approved the final draft.
- Arlina Nurhapsari conceived and designed the experiments, performed the experiments, prepared figures and/or tables, and approved the final draft.
- Adi Prayitno conceived and designed the experiments, authored or reviewed drafts of the article, and approved the final draft.
- Annisa Aghnia Rahma analyzed the data, prepared figures and/or tables, and approved the final draft.
- Muhana Fawwazy Ilyas analyzed the data, prepared figures and/or tables, and approved the final draft.

### Animal Ethics

The following information was supplied relating to ethical approvals (*i.e.*, approving body and any reference numbers):

The protocol of this study has been registered and approved by the Health Research Ethics Commission, Faculty of Medicine, Gadjah Mada University (UGM), Yogyakarta, Indonesia, with registration number KE/FK/0703/EC/2020 on 29 June 2020.

## Data Availability

The raw data are available in the Supplemental Files.

## Supplemental Information

Supplemental information for this article can be found online at http://dx.doi.org/10.7717/peerj.18004#supplemental-information.

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
