# Peer review of "Asiatic acid reduces lipopolysaccharides-induced pulp inflammation through activation of nuclear factor erythroid 2-related factor 2 in rats"

_PeerJ, doi:10.7717/peerj.18004_

## Round 0.1 · original submission · Major Revisions

Please thoroughly review and provide detailed comments on all the concerns raised by the reviewers. The work and conclusions would be greatly improved by including an analysis of the expression of pro-inflammatory mediators such as interleukin-6 (IL-6), inducible nitric oxide synthase (iNOS), tumor necrosis factor-alpha (TNF-α), and cyclooxygenase-2 (COX-2) in LPS-exposed cells. Thanks!

Reviewer 1 ·

Basic reporting

The manuscript titled Asian acid reduces lipopolysaccharides-induced pulp inflammation through activation of nuclear factor erythroid 2-related factor 2, with details, is well presented and well developed experimentally. However, it has deficiencies that make it an incomplete study.

Experimental design

The methodological part is well developed for the proposed objectives, would you only observe why use a single measurement time (72 h)?

Validity of the findings

In my opinion, the results were expected given that asiatic acid had previously demonstrated its anti-inflammatory and antioxidant activity, so it was highly predictable that it would show the induction of the expression of the nuclear marker erythroid factor 2-related factor 2 (Nrf2). ), the work and conclusions would be complete if the expression of pro-inflammatory mediators such as interleukin-6 (IL-6), inducible nitric-oxide synthase (iNOS), tumor necrosis factor-α (TNF-α), and cyclooxygenase-2 (COX-2)) in LPS-exposed cells

Additional comments

A study evaluating anti-inflammatory activity such as the one presented must be broader to have complete conclusions about the effect of asiatic acid.
Asiatic acid has previously been shown to have anti-inflammatory and antioxidant activity, so it is highly predictable that it would show the induction of the expression of the nuclear marker erythroid factor 2-related factor 2 (Nrf2), which is why the main observation against it is Why was only Nrf2 evaluated? I consider that, for example, the expression of pro-inflammatory mediators such as interleukin-6 (IL-6), inducible nitric-oxide synthase (iNOS), tumor necrosis factor-α (TNF-α) should have been evaluated and cyclooxygenase-2 (COX-2)) in LPS-exposed cells.

Reviewer 2 ·

Basic reporting

In the abstract, the authors should specify which interventions were utilized since there were 6 separated groups. There were several abbreviations which appeared for the first time without explanation. The “P” value should be written in italics.
The discussion was too prolix, the authors should focused on the discussion of Nrf2, which was priority of this study. In line 291 to 292, it seemed inappropriate to state “could attenuate the inflammation induced by LPS by increasing the Nrf2 level. “, rather it should be altered as “by Nrf2 application.”
The format of the references should be unified, for instance reference 5 did not include the month, while the other references did.

Experimental design

The authors should justify why the experiment were divided into 6 groups, especially there were 3 interventional groups, the IG1, IG2 and IG3 seemed to be unnecessary, which may be combined as one or two groups.
There were only HE stainings of the histological analysis which seemed insufficient to demonstrate inflammation. Immunohistochemical staining should also be included, especially for the detection of inflammatory factors, such as TNF-alpha, IL-1, etc. also the detection for Nrf2 should be included.

Validity of the findings

In line 182, the P value should be described as <0.01. Also was the level of TNF-alpha detected from the pulp tissue acquired? The authors should specify this.

Additional comments

A diagram or operation photograph for the animal model seemed to be necessary.

---

## Round 0.2 · Minor Revisions

The attached PDF proposes some editorial suggestions. Please consider them in a revised version that addresses grammar and style.

Reviewer 2 ·

Basic reporting

no comment

Experimental design

no comment

Validity of the findings

no comment

Additional comments

no comment

---

## Round 0.3 · accepted · Accept

Thank you for making the final minor revisions. Your manuscript has been accepted in PeerJ.